# WOR and $p$'s:
# Sketches for $\ell_p$-Sampling Without Replacement

**Edith Cohen**
Google Research
Tel Aviv University
edith@cohenwang.com

**Rasmus Pagh**
IT University of Copenhagen
BARC
Google Research
pagh@itu.dk

**David P. Woodruff**
CMU
dwoodruf@cs.cmu.edu

## Abstract

Weighted sampling is a fundamental tool in data analysis and machine learning pipelines. Samples are used for efficient estimation of statistics or as sparse representations of the data. When weight distributions are skewed, as is often the case in practice, without-replacement (WOR) sampling is much more effective than with-replacement (WR) sampling: it provides a broader representation and higher accuracy for the same number of samples. We design novel composable sketches for WOR $\ell_p$ *sampling*, weighted sampling of keys according to a power $p \in [0, 2]$ of their frequency (or for signed data, sum of updates). Our sketches have size that grows only linearly with the sample size. Our design is simple and practical, despite intricate analysis, and based on off-the-shelf use of widely implemented heavy hitters sketches such as `CountSketch`. Our method is the first to provide WOR sampling in the important regime of $p > 1$ and the first to handle signed updates for $p > 0$.

## 1 Introduction

Weighted random sampling is a fundamental tool that is pervasive in machine learning and data analysis pipelines. A sample serves as a sparse summary of the data and provides efficient estimation of statistics and aggregates.

We consider data $\mathcal{E}$ presented as *elements* in the form of key value pairs $e = (e.\mathsf{key}, e.\mathsf{val})$. We operate with respect to the *aggregated* form of keys and their *frequencies* $\nu_x := \sum_{e|e.\mathsf{key}=x} e.\mathsf{val}$, defined as the sum of values of elements with key $x$. Examples of such data sets are stochastic gradient updates (keys are parameters and element values are signed and the aggregated form is the combined gradient), search (keys are queries, elements have unit values, and the aggregated form are query-frequency pairs), or training examples for language models (keys are co-occurring terms).

The data is commonly distributed across servers or devices or is streamed and the number of distinct keys is very large. In this scenario it is beneficial to perform computations without explicitly producing a table of key-frequency pairs, as this requires storage or communication that grows linearly with the number of keys. Instead, we use *composable sketches* which are data structures that support (i) *processing a new element* $e$: Computing a sketch of $\mathcal{E} \cup \{e\}$ from a sketch of $\mathcal{E}$ and $e$ (ii) *merging*: Computing a sketch of $\mathcal{E}_1 \cup \mathcal{E}_2$ from sketches of each $\mathcal{E}_i$ and (iii) are such that the desired output can be produced from the sketch. Composability facilitates parallel, distributed, or streaming computation. We aim to design sketches of small size, because the sketch size determines the storage and communication requirements. For sampling, we aim for the sketch size to be not much larger than the desired sample size.

**The case for $p$'s:**   Aggregation and statistics of functions of the frequencies are essential for resource allocation, planning, and management of large scale systems across application areas. The need for efficiency prompted rich theoretical and applied work on streaming and sketching methods that spanned decades [57, 39, 3, 36, 41, 34, 33, 52, 51]. We study $\ell_p$ *sampling*, which refers to weighted sampling of keys with respect to a power $p$ of their frequency $\nu_x^p$. These samples support estimates of frequency statistics of the general form $\sum_x f(\nu_x) L_x$ for functions of frequency $f$ and constitute sparse representations of the data. Low powers ($p < 1$) are used to mitigate frequent keys and obtain a better resolution of the tail whereas higher powers ($p > 1$) emphasize more frequent keys. Moreover, recent work suggests that on realistic distributions, $\ell_p$ samples for $p \in [0, 2]$ provide accurate estimates for a surprisingly broad set of tasks [32].

Sampling is at the heart of stochastic optimization. When training data is distributed [54], sampling can facilitate efficient example selection for training and efficient communication of gradient updates of model parameters. Training examples are commonly weighted by a function of their frequency: Language models [56, 62] use low powers $p < 1$ of frequency to mitigate the impact of frequent examples. More generally, the function of frequency can be adjusted in the course of training to shift focus to rarer and harder examples as training progresses [8]. A sample of examples can be used to produce stochastic gradients or evaluate loss on domains of examples (expressed as frequency statistics). In distributed learning, the communication of dense gradient updates can be a bottleneck, prompting the development of methods that sparsify communication while retaining accuracy [54, 1, 66, 45]. Weighted sampling by the $p$-th powers of magnitudes complements existing methods that sparsify using heavy hitters (or other methods, e.g., sparsify randomly), provides adjustable emphasis to larger magnitudes, and retains sparsity as updates are composed.

**The case for WOR:**   Weighted sampling is classically considered with (WR) or without (WOR) replacement. We study here the WOR setting. The benefits of WOR sampling were noted in very early work [42, 40, 67] and are becoming more apparent with modern applications and the typical skewed distributions of massive datasets. WOR sampling provides a broader representation and more accurate estimates, with tail norms replacing full norms in error bounds. Figure 1 illustrates these benefits of WOR for Zipfian distributions with $\ell_1$ sampling (weighted by frequencies) and $\ell_2$ sampling (weighted by the squares of frequencies). We can see that WR samples have a smaller effective sample size than WOR (due to high multiplicity of heavy keys) and that while both WR and WOR well-approximate the frequency distribution on heavy keys, WOR provides a much better approximation of the tail.

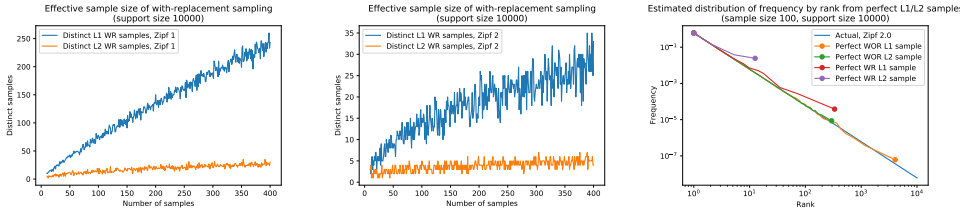

Figure 1: WOR vs WR. Left and middle: Effective vs actual sample size $\mathsf{Zipf}[\alpha = 1]$ and $\mathsf{Zipf}[\alpha = 2]$, with each point reflecting a single sample. Right: Estimates of the frequency distribution $\mathsf{Zipf}[\alpha = 2]$.

**Related work.**   The sampling literature offers many WOR sampling schemes for aggregated data: [63, 67, 12, 64, 60, 35, 23, 24, 21]. A particularly appealing technique is bottom-$k$ (order) sampling, where weights are scaled by random variables and the sample is the set of keys with top-$k$ transformed values [64, 60, 35, 23, 24]. There is also a large body of work on sketches for sampling unaggregated data by functions of frequency. There are two primary approaches. The first approach involves transforming data elements so that a bottom-$k$ sample by function of frequency is converted to an easier problem of finding the top-$k$ keys sorted according to the *maximum* value of an element with the key. This approach yields WOR distinct ($\ell_0$) sampling [50], $\ell_1$ sampling [39, 20], and sampling with respect to any concave sublinear functions of frequency (including $\ell_p$ sampling for $p \leq 1$) [18, 22]). These sketches work with non-negative element values but only provide limited support for negative updates [38, 20]. The second approach performs WR $\ell_p$ sampling for $p \in [0, 2]$ using sketches that are random projections [43, 37, 4, 48, 58, 5, 49, 47]. The methods support signed

updates but were not adapted to WOR sampling. For $p > 2$, a classic lower bound [3, 6] establishes that sketches of size polynomial in the number of distinct keys are required for worst case frequency distributions. This task has also been studied in distributed settings [14, 46]; [46] observes the importance of WOR in that setting though does not allow for updates to element values.

---

**Algorithm 1:** WORp (high level)

---

**Components:**
    Random hash map $r_x \sim \mathsf{Exp}[1]$                           `// Map keys `$x$` to i.i.d `$r_x$
    $\ell_q$ residual Heavy Hitters (rHH) method
**Input:** Data elements $\mathcal{E}$ as key value pairs $e = (e.\mathsf{key}, e.\mathsf{val})$
    $p \in (0, 2]$                              `// Specifies `$p \le q$` for `$\ell_p$` sampling`
    $k \ge 1$                                       `// Sample Size`
    Specify one or two passes `// One-pass requires a more accurate/larger sketch`
**Initialization:**
    Initialize rHH sketch $R$            `// Size determined by `$p$`, `$k$`, one/two passes`
**Process data element** $e = (e.\mathsf{key}, e.\mathsf{val})$:
    $R.\mathtt{Process}(e.\mathsf{key}, e.\mathsf{val}/r_{e.\mathsf{key}}^{1/p})$   `// Transform element and insert into sketch`
**Final:**
    Extract sample from the sketch $R$ `// directly (one pass) or with a second pass.`

---

**Contributions:** We present WORp: A method for WOR $\ell_p$ sampling for $p \in [0, 2]$ via composable sketches of size that grows linearly with the sample size (see pseudocode in Algorithm 1). WORp is simple and practical and uses a bottom-$k$ transform (see Figure 2) to reduce sampling to a top-$k$ problem on the transformed data. The technical heart of the paper is establishing that for any set of input frequencies, the keys with top-$k$ transformed frequencies are *(residual) heavy hitters* (rHH) and therefore can be identified using a small sketch. In terms of implementation, WORp only requires an off-the-shelf use of popular (and widely-implemented) HH sketches [57, 53, 13, 33, 55, 9]. WORp is the first WOR $\ell_p$ sampling method (that uses sample-sized sketches) for the regime $p \in (1, 2]$ and the first to fully support negative updates for $p \in (0, 2]$. As a bonus, we include practical optimizations (that preserve the theoretical guarantees) and perform experiments that demonstrate both the practicality and accuracy of WORp.[1]

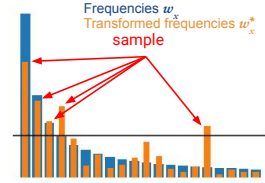

Frequencies $w_x$
Transformed frequencies $w_x^*$
sample

In addition to the above, we show that perhaps surprisingly, it is possible to obtain a WOR $\ell_p$-sample of a set of $k$ indices, for any $p \in [0, 2]$, with variation distance at most $\frac{1}{\mathrm{poly}(n)}$ to a true WOR $\ell_p$-sample, and using only $k \cdot \mathrm{poly}(\log n)$ bits of memory. Our variation distance is extremely small, and cannot be detected by any polynomial time algorithm. This makes it applicable in settings for which privacy may be a concern; indeed, this shows that no polynomial time algorithm can learn anything from the sampled output other than what follows from a simulator who outputs a WOR $\ell_p$-sample from the actual (variation distance 0) distribution. Finally, for $p \in (0, 2)$, we show that the memory of our algorithm is optimal up to an $O(\log^2 \log n)$ factor.

Figure 2: Illustration of bottom-$k$ sampling.

## 2 Preliminaries

A dataset $\mathcal{E}$ consists of *data elements* that are key value pairs $e = (e.\mathsf{key}, e.\mathsf{val})$. The frequency of a key $x$, denoted $\nu_x := \sum_{e|e.\mathsf{key}=x} e.\mathsf{val}$, is the sum of values of elements with key $x$. We use the notation $\boldsymbol{\nu}$ for a vector of frequencies of keys.

For a function $f$ and vector $\boldsymbol{w}$, we denote the vector with entries $f(w_x)$ by $f(\boldsymbol{w})$. In particular, $\boldsymbol{w}^p$ is the vector with entries $w_x^p$ that are the $p$-th powers of the entries of $\boldsymbol{w}$. For vector $\boldsymbol{w} \in \Re^n$ and index $i$, we denote by $w_{(i)}$ the value of the entry with the $i$-th largest magnitude in $\boldsymbol{w}$. We denote by $\mathsf{order}(\boldsymbol{w})$ the permutation of the indices $[n] = \{1, 2, \ldots, n\}$ that corresponds to decreasing order of entries by magnitude. For $k \ge 1$, we denote by $\mathsf{tail}_k(\boldsymbol{w})$ the vector with the $k$ entries with largest magnitudes removed (or replaced with 0).

In the remainder of the section we review ingredients that we build on: bottom-$k$ sampling, implementing a bottom-$k$ transform on unaggregated data, and composable sketch structures for residual heavy hitters (rHH).

## 2.1 Bottom-$k$ sampling (ppswor and priority)

Bottom-$k$ sampling (also called order sampling [64]) is a family of without-replacement weighted sampling schemes of a set $\{(x, w_x)\}$ of key and weight pairs. The weights $(x, w_x)$ are transformed via independent random maps $w_x^T \leftarrow \frac{w_x}{r_x}$, where $r_x \sim \mathcal{D}$ are i.i.d. from some distribution $\mathcal{D}$. The sample includes the pairs $(x, w_x)$ for keys $x$ that are top-$k$ by transformed magnitudes $|\boldsymbol{w^T}|$ [59, 64, 35, 15, 11, 23, 25] [2]. For estimation tasks, we also include a *threshold* $\tau := |w_{(k+1)}^T|$, the $(k+1)$-st largest magnitude of transformed weights. Bottom-$k$ schemes differ by the choice of distribution $\mathcal{D}$. Two popular choices are Probability Proportional to Size WithOut Replacement (ppswor) [64, 15, 25] via the exponential distribution $\mathcal{D} \leftarrow \mathsf{Exp}[1]$ and Priority (Sequential Poisson) sampling [59, 35] via the uniform distribution $\mathcal{D} \leftarrow U[0, 1]$. Ppswor is equivalent to a weighted sampling process [63] where keys are drawn successively (without replacement) with probability proportional to their weight. Priority sampling mimics a pure Probability Proportional to Size (pps) sampling, where sampling probabilities are proportional to weights (but truncated to be at most 1).

**Estimating statistics from a Bottom-$k$ sample.** Bottom-$k$ samples provide us with unbiased inverse-probability [42] per-key estimates on $f(w_x)$, where $f$ is a function applied to the weight [2, 24, 19, 17]):

$$\widehat{f(w_x)} := \begin{cases} \frac{f(w_x)}{\Pr_{r \sim \mathcal{D}}[r \leq |w_x|/\tau]} & \text{if } x \in S \\ 0 & \text{if } x \notin S \end{cases} . \tag{1}$$

These estimates can be used to sparsify a vector $f(\boldsymbol{w})$ to $k$ entries or to estimate sum statistics of the general form:

$$\sum_x f(w_x) L_x \tag{2}$$

using the unbiased estimate

$$\widehat{\sum_x f(w_x) L_x} := \sum_x \widehat{f(w_x)} L_x = \sum_{x \in S} \widehat{f(w_x)} L_x .$$

The quality of estimates depends on $f$ and $L$. We measure the quality of these unbiased estimates by the sum over keys of the per-key variance. With both ppswor and priority sampling and $f(w) := w$, the sum is bounded by a respective one for pps with replacement. The per-key variance bound is

$$\mathsf{Var}[\widehat{w_x}] \leq \frac{1}{k-1} w_x \|\boldsymbol{w}\|_1 \tag{3}$$

and the respective sum by $\sum_x \mathsf{Var}[\widehat{w_x}] \leq \frac{1}{k-1} \|\boldsymbol{w}\|_1^2$. This can be tightened to $\mathsf{Var}[\widehat{w_x}] \leq \min\{O(\frac{1}{k}) w_x \|\mathsf{tail}_k(\boldsymbol{w})\|_1, \exp\left(-O(k)\frac{w_x}{\|\mathsf{tail}_k(\boldsymbol{w})\|_1}\right) w_x^2\}$ and respective bound on the sum of $O(\|\mathsf{tail}_k(\boldsymbol{w})\|_1^2/k)$. For skewed distributions, $\|\mathsf{tail}_k(\boldsymbol{w})\|_1^2 \ll \|\boldsymbol{w}\|_1^2$ and hence WOR sampling is beneficial. Conveniently, bottom-$k$ estimates for different keys $x_1 \neq x_2$ have non-positive correlations $\mathsf{Cov}[\widehat{w_{x_1}}, \widehat{w_{x_2}}] \leq 0$, so the variance of sum estimates is at most the respective weighted sum of per-key variance. Note that the per-key variance for a function of weight is $\mathsf{Var}[\widehat{f(w_x)}] = \frac{f(w_x)^2}{w_x^2} \mathsf{Var}[\widehat{w_x}]$. WOR (and WR) estimates are more accurate (in terms of normalized variance sum) when the sampling is weighted by $f(\boldsymbol{w})$.

## 2.2 Bottom-$k$ sampling by power of frequency

To perform bottom-$k$ sampling of $\boldsymbol{w}^p$ with distribution $\mathcal{D}$, we draw $r_x \sim \mathcal{D}$, transform the weights $w_x^T \leftarrow w_x^p/r_x$, and return the top-$k$ keys in $\boldsymbol{w}^T$. This is equivalent to bottom-$k$ sampling the vector

$w$ using the distribution $\mathcal{D}^{1/p}$, that is, draw $r_x \sim \mathcal{D}$, transform the weights

$$w_x^* \leftarrow \frac{w_x}{r_x^{1/p}} \tag{4}$$

and return the top-$k$ keys according to $w^*$. Equivalence is because $(w_x^*)^p = \left(\frac{w_x}{r_x^{1/p}}\right)^p = \frac{w_x^p}{r_x} = w_x^T$ and $f(x) = x^p$ is a monotone increasing and hence $\mathsf{order}(w^*) = \mathsf{order}(w^T)$. We denote the distribution of $w^*$ obtained from the bottom-$k$ transform (4) as $p\text{-}\mathcal{D}[w]$ and specifically, $p\text{-ppswor}[w]$ when $\mathcal{D} = \mathsf{Exp}[1]$ and $p\text{-priority}[w]$ when $\mathcal{D} = U[0,1]$. We use the term $p$-ppswor for bottom-$k$ sampling by $\mathsf{Exp}^{1/p}$.

The linear transform (4) can be efficiently performed over unaggregated data by using a random hash to represent $r_x$ for keys $x$ and then locally generating an output element for each input element

$$(e.\mathsf{key}, e.\mathsf{val}) \rightarrow (e.\mathsf{key}, e.\mathsf{val}/r_{e.\mathsf{key}}^{1/p}) \tag{5}$$

The task of sampling by $p$-th power of frequency $\nu^p$ is replaced by the task of top-$k$ by frequency $\nu_x^* := \sum_{e \in \mathcal{E}^* | e.\mathsf{key}=x} e.\mathsf{val}$ on the respective output dataset $\mathcal{E}^*$, noting that $\nu_x^* = \nu_x/r_x^{1/p}$. Therefore, the top-$k$ keys in $\nu^*$ are a bottom-$k$ sample according to $\mathcal{D}$ of $\nu^p$. Note that we can approximate the input frequency $\nu_x'$ of a key $x$ from an approximate output frequency $\widehat{\nu_x^*}$ using the hash $r_x$. Note that relative error is preserved:

$$\nu_x' \leftarrow \widehat{\nu_x^*} r_x^{1/p} \ . \tag{6}$$

This per-element scaling was proposed in the *precision sampling* framework of Andoni et al. [5] and inspired by a technique for frequency moment estimation using stable distributions [43].

Generally, finding the top-$k$ frequencies is a task that requires large sketch structures of size linear in the number of keys. However, [5] showed that when the frequencies are drawn from $p\text{-priority}[w]$ (applied to arbitrary $w$) and $p \leq 2$ then the top-1 value is likely to be an $\ell_2$ *heavy hitter*. Here we refine the analysis and use the more subtle notion of *residual heavy hitters* [9]. We show that the top-$k$ output frequencies in $w^* \sim p\text{-ppswor}[w]$ are very likely to be $\ell_q$ residual heavy hitters (when $q \geq p$) and can be found with a sketch of size $\tilde{O}(k)$.

## 2.3   Residual heavy hitters (rHH)

An entry in a weight vector $w$ is called an $\varepsilon\text{-}$ *heavy hitter* if $w_x \geq \varepsilon \sum_y w_y$. A heavy hitter with respect to a function $f$ is defined as a key with $f(w_x) \geq \varepsilon \sum_y f(w_y)$. When $f(w) = w^q$, we refer to a key as an $\ell_q$ heavy hitter. For $k \geq 1$ and $\psi > 0$, a vector $w$ has $(k, \psi)$ *residual heavy hitters* [9] when the top-$k$ keys are "heavy" with respect to the tail of the frequencies starting at the $(k+1)$-st most frequent key, that is, $\forall i \leq k, \ w_{(i)} \geq \frac{\psi}{k} \|\mathsf{tail}_k(w)\|_1$. This is equivalent to $\frac{\|\mathsf{tail}_k(w)\|_1}{w_{(k)}} \leq \frac{k}{\psi}$. We say that $w$ has rHH with respect to a function $f$ if $f(w)$ has rHH. In particular, $w$ has $\ell_q$ $(k, \psi)$ rHH if

$$\frac{\|\mathsf{tail}_k(w)\|_q^q}{w_{(k)}^q} \leq \frac{k}{\psi} \ . \tag{7}$$

Popular composable HH sketches were adopted to rHH and include (see Table 1): (i) $\ell_1$ sketches designed for positive data elements. A deterministic counter-based variety [57, 53, 55] with rHH adaptation [9] and the randomized $\mathtt{CountMin}$ sketch [33]. (ii) $\ell_2$ sketches designed for signed data elements, notably $\mathtt{CountSketch}$ [13] with rHH analysis in [49]. With these designs, a sketch for $\ell_q$ $(k, \psi)$-rHH provides estimates $\widehat{\nu_x}$ for all keys $x$ with error bound:

$$\|\widehat{\nu} - \nu\|_\infty^q \leq \frac{\psi}{k} \|\mathsf{tail}_k(\nu)\|_q^q \ . \tag{8}$$

With randomized sketches, the error bound (8) is guaranteed with some probability $1 - \delta$. $\mathtt{CountSketch}$ has the advantages of capturing top keys that are $\ell_2$ but not $\ell_1$ heavy hitters and supports signed data, but otherwise provides lower accuracy than $\ell_1$ sketches for the same sketch size. Methods also vary in supported key domains: counters natively work with key strings whereas randomized sketches work for keys from $[n]$ (see Appendix **??** for a further discussion). We use these sketches off-the-shelf through the following operations:

- $R.\texttt{Initialize}(k, \psi, \delta)$: Initialize a sketch structure
- $\texttt{Merge}(R_1, R_2)$: Merge two sketches with the same parameters and internal randomization
- $R.\texttt{Process}(e)$: process a data element $e$
- $R.\texttt{Est}(x)$: Return an estimate of the frequency of a key $x$.

| Sketch ($\ell_q$, sign) | Size | $\\|\widehat{\boldsymbol{\nu}} - \boldsymbol{\nu}\\|_\infty^q \leq$ |
|---|---|---|
| $\texttt{Counters}$ ($\ell_1$, +) [9] | $O(\frac{k}{\psi})$ | $\frac{\psi}{k}\\|\mathsf{tail}_k(\boldsymbol{\nu})\\|_1$ |
| $\texttt{CountSketch}$ ($\ell_2$, ±) [13] | $O(\frac{k}{\psi}\log\frac{n}{\delta})$ | $\frac{\psi}{k}\\|\mathsf{tail}_k(\boldsymbol{\nu})\\|_2^2$ |

Table 1: Sketches for $\ell_q$ $(k, \psi)$ rHH.

## 3 WORp Overview

Our WORp methods apply a $p$-ppswor transform to data elements (5) and (for $q \geq p$) compute an $\ell_q$ $(k, \psi)$-rHH sketch of the output elements. The rHH sketch is used to produce a sample of $k$ keys.

We would like to set $\psi$ to be low enough so that for any input frequencies $\boldsymbol{\nu}$, the top-$k$ keys by transformed frequencies $\boldsymbol{\nu}^* \sim p\text{-ppswor}[\boldsymbol{\nu}]$ are rHH (satisfy condition (7)) with probability at least $1 - \delta$. We refine this desiderata to be conditioned on the permutation of keys in $\mathsf{order}(\boldsymbol{\nu}^*)$. This conditioning turns out not to further constrain $\psi$ and allows us to provide the success probability uniformly for any potential $k$-sample. Since our sketch size grows inversely with $\psi$ (see Table 1), we want to use the maximum value that works. We will be guided by the following:

$$\Psi_{n,k,\rho=q/p}(\delta) := \sup\left\{\psi \mid \forall \boldsymbol{w} \in \Re^n, \pi \in S^n \underset{\boldsymbol{w}^* \sim p\text{-ppswor}[\boldsymbol{w}]|\mathsf{order}(\boldsymbol{w}^*)=\pi}{\Pr}\left[k\frac{|w_{(k)}^*|^q}{\\|\mathsf{tail}_k(\boldsymbol{w}^*)\\|_q^q} \leq \psi\right] \leq \delta\right\},$$
(9)

where $S^n$ denotes the set of permutations of $[n]$. If we set the rHH sketch parameter to $\psi \leftarrow \varepsilon \Psi_{n,k,\rho}$ then using (8), with probability at least $1 - \delta$,

$$\\|\widehat{\boldsymbol{\nu}^*} - \boldsymbol{\nu}^*\\|_\infty^q \leq \frac{\psi}{k}\\|\mathsf{tail}_k(\boldsymbol{\nu}^*)\\|_q^q = \varepsilon\frac{\Psi_{n,k,\rho}(\lambda)}{k}\\|\mathsf{tail}_k(\boldsymbol{\nu}^*)\\|_q^q \leq \varepsilon|\nu_{(k)}^*|^q .$$
(10)

We establish the following lower bounds on $\Psi_{n,k,\rho}(\delta)$:

**Theorem 3.1.** *There is a universal constant $C > 0$ such that for all $n$, $k > 1$, and $\rho = q/p$*

$$\text{For } \rho = 1\text{: } \Psi_{n,k,\rho}(3e^{-k}) \geq \frac{1}{C\ln\frac{n}{k}}$$
(11)

$$\text{For } \rho > 1\text{: } \Psi_{n,k,\rho}(3e^{-k}) \geq \frac{1}{C}\max\{\rho - 1, \frac{1}{\ln\frac{n}{k}}\} .$$
(12)

To set sketch parameters in implementations, we approximate $\Psi$ using simulations of what we establish to be a "worst case" frequency distribution. For this we use a specification of a "worst-case" set of frequencies as part of the proof of Theorem 3.1 (See the full version). From simulations we obtain that very small values of $C < 2$ suffice for $\delta = 0.01$, $\rho \in \{1, 2\}$, and $k \geq 10$.

We analyse a few WORp variants. The first we consider returns an exact $p$-ppswor sample, including exact frequencies of keys, using two passes. We then consider a variant that returns an approximate $p$-ppswor sample in a single pass. The two-pass method uses smaller rHH sketches and efficiently works with keys that are arbitrary strings.

We also provide another rHH-based technique that provides a guaranteed very small variation distance on $k$-tuples in a single pass.

## 4 Two-pass WORp

We design a two-pass method for ppswor sampling according to $\boldsymbol{\nu}^p$ for $p \in (0, 2]$ (Equivalently, a $p$-ppswor sample according to $\boldsymbol{\nu}$):

| sign, $p$ | Sketch size words | key strings | Pr[success] |
|---|---|---|---|
| $\pm, p < 2$ | $O(k \log n)$ | $O(k)$ | $(1 - \frac{1}{\text{poly}(n)})(1 - 3e^{-k})$ |
| $\pm, p = 2$ | $O(k \log^2 n)$ | $O(k)$ | $(1 - \frac{1}{\text{poly}(n)})(1 - 3e^{-k})$ |
| $+, p < 1$ | $O(k)$ | $O(k)$ | $1 - 3e^{-k}$ |
| $+, p = 1$ | $O(k \log n)$ | $O(k)$ | $1 - 3e^{-k}$ |

Table 2: Two-pass ppswor sampling of $k$ keys according to $\boldsymbol{\nu}^p$. Sketch size (memory words and number of stored key strings). For $p \in (0, 2]$ and signed ($\pm$) or positive ($+$) value elements.

- **Pass I:** We compute an $\ell_q$ $(k + 1, \psi)$-rHH sketch $R$ of the transformed data elements

$$(\texttt{KeyHash}(e.\mathsf{key}), e.\mathsf{val}/r_{e.\mathsf{key}}^{1/p}) \ . \tag{13}$$

A hash $\texttt{KeyHash} \rightarrow [n]$ is applied when keys have a large or non-integer domain to facilitate use of $\texttt{CountSketch}$ or reduce storage with $\texttt{Counters}$. We set $\psi \leftarrow \frac{1}{3^q} \Psi_{n,k,\rho}(\delta)$.

- **Pass II:** We collect key strings $x$ (if needed) and corresponding exact frequencies $\nu_x$ for keys with the $Bk$ largest $|\widehat{\nu_x^*}|$, where $B$ is a constant (see below) and $\widehat{\nu_x^*} := R.\texttt{Est}[\texttt{KeyHash}(x)]$ are the estimates of $\nu_x^*$ provided by $R$. For this purpose we use a composable top-$(Bk)$ sketch structure $T$. The size of $T$ is dominated by storing $Bk$ key strings.

- **Producing a $p$-ppswor sample from $T$:** Compute exact transformed frequencies $\nu_x^* \leftarrow \nu_x r_x^{1/p}$ for stored keys $x \in T$. Our sample is the set of key frequency pairs $(x, \nu_x)$ for the top-$k$ stored keys by $\nu_x^*$. The threshold $\tau$ is the $(k + 1)^{\text{th}}$ largest $\nu_x^*$ over stored keys.

- **Estimation:** We compute per-key estimates as in (1): Plugging in $\mathcal{D} = \textsf{Exp}[1]^{1/p}$ for $p$-ppswor, we have $\widehat{f(\nu_x)} := 0$ for $x \notin S$ and for $x \in S$ is $\widehat{f(\nu_x)} := \frac{f(\nu_x)}{1 - \exp\left(-(\frac{\nu_x}{\tau})^p\right)}$.

We establish that the method returns the $p$-ppswor sample with probability at least $1 - \delta$, propose practical optimizations, and analyze the sketch size:

**Theorem 4.1.** *The 2-pass method returns a $p$-ppswor sample of size $k$ according to $\boldsymbol{\nu}$ with success probability and composable sketch sizes as detailed in Table 2. The success probability is defined to be that of returning the exact top-$k$ keys by transformed frequencies. The bound applies even when conditioned on the top-$k$ being a particular $k$-tuple.*

*Proof.* From (10), the estimates $\widehat{\nu_x^*} = R.\texttt{Est}[\texttt{KeyHash}(x)]$ of $\nu_x^*$ are such that:

$$\Pr\left[\forall x \in \{e.\mathsf{key} \mid e \in \mathcal{E}\}, |\widehat{\nu_x^*} - \nu_x^*| \leq \frac{1}{3}|\nu_{(k+1)}^*|\right] \geq 1 - \delta \ . \tag{14}$$

We set $B$ in the second pass so that the following holds:

The top-$(k + 1)$ keys by $\boldsymbol{\nu}^*$ are a subset of the top-$(B(k + 1))$ keys by $\widehat{\boldsymbol{\nu}^*}$. (15)

Note that for any frequency distribution with rHH, it suffices to store $O(k/\psi)$ keys to have (15). We establish (see the appendix) that for our particular distributions, a constant $B$ suffices.

Correctness for the final sample follows from property (15) : $T$ storing the top-$(k+1)$ keys in the data according to $\boldsymbol{\nu}^*$. To conclude the proof of Theorem 4.1 we need to specify the rHH sketch structure we use. From Theorem 3.1 we obtain a lower bound on $\Psi_{n,k,\rho}$ for $\delta = 3e^{-k}$ and we use it to set $\psi$. For our rHH sketch we use $\texttt{CountSketch}$ ($q = 2$ and supports signed values) or $\texttt{Counters}$ ($q = 1$ and positive values). The top two lines in Table 2 are for $\texttt{CountSketch}$ and the next two lines are for $\texttt{Counters}$. The rHH sketch sizes follow from $\psi$ and Table 1. $\square$

## 5 One-pass WORp

Our 1-pass WORp yields a sample of size $k$ that approximates a $p$-ppswor sample of the same size and provides similar guarantees on estimation quality.

- **Sketch:** For $q \geq p$ and $\varepsilon \in (0, \frac{1}{3}]$ Compute an $\ell_q$ $(k+1, \psi)$-rHH sketch $R$ of the transformed data elements (5) where $\psi \leftarrow \varepsilon^q \Psi_{n,k+1,\rho}$.

- **Produce a sample:** Our sample $S$ includes the keys with top-$k$ estimated transformed frequencies $\widehat{\nu^*_x} := R.\text{Est}[x]$. For each key $x \in S$ we include $(x, \nu'_x)$, where the approximate frequency $\nu'_x \leftarrow \widehat{\nu^*_x} r_x^{1/p}$ is according to (6). We include with the sample the threshold $\tau \leftarrow \widehat{\nu^*}_{(k+1)}$.

- **Estimation:** We treat the sample as a $p$-ppswor sample and compute per-key estimates as in (1), substituting approximate frequencies $\nu'_x$ for actual frequencies $\nu_x$ of sampled keys and the 1-pass threshold $\widehat{\nu^*}_{(k+1)}$ for the exact $\nu^*_{(k+1)}$. The estimate is $\widehat{f(\nu_x)} := 0$ for $x \notin S$ and for $x \in S$ is:

$$\widehat{f(\nu_x)} := \frac{f(\nu'_x)}{1 - \exp\left(-(\frac{\nu'_x}{\widehat{\nu^*}_{(k+1)}})^p\right)} = \frac{f(\widehat{\nu^*_x} r_x^{1/p})}{1 - \exp\left(-r_x(\frac{\widehat{\nu^*_x}}{\widehat{\nu^*}_{(k+1)}})^p\right)} \tag{16}$$

We relate the quality of the estimates to those of a perfect $p$-ppswor. Since our 1-pass estimates are biased (unlike the unbiased perfect $p$-ppswor), we consider both bias and variance. The proof is provided in the full version.

**Theorem 5.1.** *Let $f(w)$ be such that $|f((1+\varepsilon)w) - f(w)| \leq c\varepsilon f(w)$ for some $c > 0$ and $\varepsilon \in [0, 1/2]$. Let $\widehat{f(\nu_x)}$ be per-key estimates obtained with a one-pass WORp sample and let $\widehat{f(\nu_x)}'$ be the respective estimates obtained with a (perfect) p-ppswor sample. Then $\mathsf{Bias}[\widehat{f(\nu_x)}] \leq O(\varepsilon) f(\nu_x)$ and $\mathsf{MSE}[\widehat{f(\nu_x)}] \leq (1 + O(\varepsilon)) \mathsf{Var}[\widehat{f(\nu_x)}'] + O(\epsilon) f(\nu_x)^2$.*

Note that the assumption on $f$ holds for $f(w) = w^p$ with $c = (1.5)^p$. Also note that the bias bound implies a respective contribution to the relative error of $O(\varepsilon)$ on all sum estimates.

## 6 One-pass Total Variation Distance Guarantee

We provide another 1-pass method, based on the combined use of rHH and known WR perfect $\ell_p$ sampling sketches [47] to select a $k$-tuple with a polynomially small total variation (TV) distance from the $k$-tuple distribution of a perfect $p$-ppswor. The method uses $O(k)$ (for variation distance $2^{-\Theta(k)}$, and $O(k \log n)$ for variation distance $1/n^C$ for an arbitrarily large constant $C > 0$) perfect samplers (each providing a single WR sample) and an rHH sketch. The perfect samplers are processed in sequence with prior selections "subtracted" from the linear sketch (using approximate frequencies provided by the rHH sketch) to uncover fresh samples. As with WORp, exact frequencies of sampled keys can be obtained in a second pass or approximated using larger sketches in a single pass. Details are provided in the full version.

**Theorem 6.1.** *There is a 1-pass method via composable sketches of size $O(k \operatorname{polylog}(n))$ that returns a $k$-tuple of keys such that the total variation distance from the $k$-tuples produced by a perfect $p$-ppswor sample is at most $1/n^C$ for an arbitrarily large constant $C > 0$. The method applies to keys from a domain $[n]$, and signed values with magnitudes and intermediate frequencies that are polynomial in $n$.*

We also show in the full version that our sketches in Theorem 6.1 use $O(k \cdot \log^2 n (\log \log n)^2)$ bits of memory for $0 < p < 2$, and we prove a matching lower bound on the memory required of any algorithm achieving this guarantee, up to a $(\log \log n)^2$ factor. For $p = 2$ we also show they are of optimal size, up to an $O(\log n)$ factor.

## 7 Experiments

We simulated 2-pass and 1-pass WORp in Python using `CountSketch` with 15 repetitions and table size $2k$ (total space $30k$) as our rHH sketch. Figure 3 reports estimates of the rank-frequency distribution obtained with 1-pass and 2-pass WORp and perfect WOR ($p$-ppswor) and perfect WR samples (shown for reference). For best comparison, all WOR methods use the same randomization of the $p$-ppswor transform. Table 3 reports normalized root averaged squared errors (NRMSE) on example statistics. As expected, 2-pass WORp and perfect 2-ppswor are similar and WR $\ell_2$ samples

| $\ell_p$ | $\alpha$ | $p'$ | perfect WR | perfect WOR | 1-pass WORp | 2-pass WORp |
|---|---|---|---|---|---|---|
| $\ell_2$ | Zipf[2] | $\nu^3$ | 1.16e-04 | 2.09e-11 | 1.06e-03 | 2.08e-11 |
| $\ell_2$ | Zipf[2] | $\nu^2$ | 7.96e-05 | 1.26e-07 | 1.14e-02 | 1.25e-07 |
| $\ell_1$ | Zipf[2] | $\nu$ | 9.51e-03 | 1.60e-03 | 2.79e-02 | 1.60e-03 |
| $\ell_1$ | Zipf[1] | $\nu^3$ | 3.59e-01 | 5.73e-03 | 5.14e-03 | 5.72e-03 |
| $\ell_1$ | Zipf[2] | $\nu^3$ | 3.45e-04 | 7.34e-10 | 5.11e-05 | 7.38e-10 |

Table 3: NRMSE on estimates of frequency moments on statistics of the form $\|\boldsymbol{\nu}\|_{p'}^{p'}$ from $\ell_p$ samples ($p = 1, 2$). Zipf[$\alpha$] distributions with support size $n = 10^4$, $k = 100$ samples, averaged over 100 runs. `CountSketch` of size $2k \times 31$

are less accurate with larger skew or on the tail. Note that current state of the art sketching methods are not more efficient for WR sampling than for WOR sampling, and estimation quality with perfect methods is only shown for reference. We can also see that 1-pass WORp performs well, although it requires more accuracy (lager sketch size) since it works with estimated weights (reported results are with fixed `CountSketch` size of $k \times 31$ for all methods).

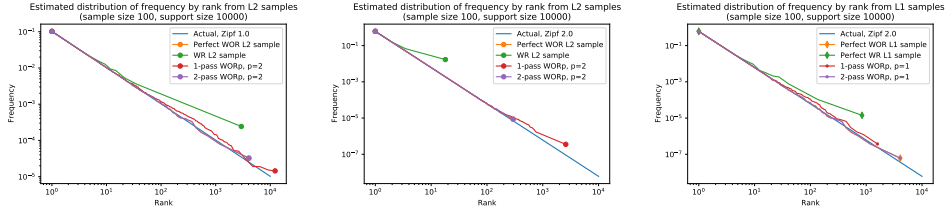

Figure 3: Estimates of the rank-frequency distribution of Zipf[1] and Zipf[2]. Using WORp 1-pass, WORp 2-pass with `CountSketch` (matrix $k \times 31$), perfect WOR, and perfect WR. Estimates from a (representative) single sample of size $k = 100$. Left and Center: $\ell_2$ sampling. Right: $\ell_1$ sampling.

**Conclusion**

We present novel composable sketches for without-replacement (WOR) $\ell_p$ sampling, based on "reducing" the sampling problem to a heavy hitters (HH) problem. The reduction, which is simple and practical, allows us to use existing implementations of HH sketches to perform WOR sampling. Moreover, streaming HH sketches that support time decay (for example, sliding windows [7]) provide a respective time-decay variant of sampling. We present two approaches, WORp, based on a bottom-$k$ transform, and another technique based on "perfect" with-replacement sampling sketches, which provides 1-pass WOR samples with negligible variation distance to a true sample. Our methods open the door for a wide range of future applications: In particular, WORp provides efficient *coordinated* bottom-$k$ samples (aka bottom-$k$ sketches) of datasets. WORp produces bottom-$k$ samples with respect to a specified randomization $r_x$ over the support (with 1-pass WORp we obtain approximate bottom-$k$ samples). Samples of different datasets or different $p$ values or different time-decay functions that are generated with the same $r_x$ are coordinated [10, 65, 61, 15, 30, 11]. Coordination is a desirable and powerful property: Samples are locally sensitivity (LSH) and change little with small changes in the dataset [10, 65, 44, 31, 17]. This allows for a compact representation of multiple samples, efficient updating, and sketch-based similarity searches. Moreover, coordinated samples (sketches) facilitate powerful estimators for multi-set statistics and similarity measures such as weighted Jaccard similarity, min or max sums, and 1-sided distance norms [15, 11, 26, 29, 27, 28, 16].

## 8 Broader Impact

Broader Impact Discussion is not applicable. We presented a method for WOR sampling that has broad applications in ML. But this is a technical paper with no particular societal implications.

## 9 Funding disclosure

D. Woodruff would like to thank for partial support from the Office of Naval Research (ONR) grant N00014-18-1-2562, the National Science Foundation (NSF) under Grant No. CCF-1815840, and a Simons Investigator Award. R. Pagh has been supported by Villum Foundation grant 16582 to Basic Algorithms Research Copenhagen (BARC) and European Research Council / ERC grant agreement no. 614331.

## Footnotes

[1]Code for the experiments is provided in the following Colab notebook `https://colab.research.google.com/drive/1Tix7SwsPp7A_OtSuaRf3IwfTH-qo9_81?usp=sharing`

[2]Historically, the term bottom-$k$ is due to analogous use of $1/w_x^T$, but here we find it more convenient to work with "top-$k$"

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
