[Supplementary Material]

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

**Lemma 4.1.** *A sufficient condition for property* (15) *is that* $|\nu_{(B(k+1))}^*| \leq \frac{1}{3}|\nu_{((k+1))}^*|$.

*Proof.* Note that $|\widehat{\nu_x^*}| \geq \frac{2}{3}|\nu_{(k+1)}^*|$ for keys that are top-$(k + 1)$ by $\boldsymbol{\nu}^*$ and $|\widehat{\nu^*}_{(B(k+1))}| \leq |\nu_{(B(k+1))}^*| + \frac{1}{3}|\nu^*(k+1)|$. Hence $|\widehat{\nu_x^*}| \geq |\widehat{\nu^*}_{(B(k+1))}|$ for all keys that are top-$(k + 1)$ by $\boldsymbol{\nu}^*$. $\square$

We then use Lemma E.1 to express a "worst-case" distribution for the ratio $\nu_{B(k+1)}^*/\nu_{(}^* k + 1)$ and use the latter (using Corollary D.2) to show that the setting of $\Psi(\delta)$ according to our proof of Theorem 3.1 (Appendix B-D) implies that the conditions that guarantee the rHH property will also imply a ratio of at most $1/3$ with a constant $B$.

Correctness for the final sample follows from property (15): $T$ storing the top-$(k+1)$ keys in the data according to $\boldsymbol{\nu}^*$. To conclude the proof of Theorem 4.1 we need to specify the rHH sketch structure we use. From Theorem 3.1 we obtain a lower bound on $\Psi_{n,k,\rho}$ for $\delta = 3e^{-k}$ and we use it to set $\psi$. For our rHH sketch we use `CountSketch` ($q=2$ and supports signed values) or `Counters` ($q=1$ and positive values). The top two lines in Table 2 are for `CountSketch` and the next two lines are for `Counters`. The rHH sketch sizes follow from $\psi$ and Table 1. □

### 4.1 Practical optimizations

We propose practical optimizations that improve efficiency without compromising quality guarantees.

The first optimization allows us to store $k' \ll B(k+1)$ keys in the second pass: We always store the top-$(k+1)$ keys by $\widehat{\boldsymbol{\nu}^*}$ but beyond that only store keys if they satisfy

$$\widehat{\nu^*}_x \geq \frac{1}{2}\widehat{\nu^*}_{(k+1)} \ , \tag{16}$$

where $\boldsymbol{\nu}^*$ is with respect to data elements processed by the current sketch. We establish correctness:

**Lemma 4.2.** *1. There is a composable structure that only stores keys that satisfy (16) and collects exact frequencies for these keys.*

*2. If (14) holds, the top-$(k+1)$ keys by $\boldsymbol{\nu}^*$ satisfy (16) (and hence will be stored in the sketch).*

*Proof.* (i) The structure is a slight modification of a top-$k'$ structure. Since $\widehat{\nu^*}_{(k+1)}$ can only increase as more distinct keys are processed, the condition (16) only becomes more stringent as we merge sketches and process elements. Therefore if a key satisfies the condition at some point it would have satisfied the condition when elements with the key were previously processed and therefore we can collect exact frequencies.

(ii) From the uniform approximation (14), we have $\widehat{\nu^*}_{(k+1)} \leq \frac{4}{3}\nu^*_{(k+1)}$. Let $x$ be the $(k+1)$-th key by $|\boldsymbol{\nu}^*|$. Its estimated transformed frequency is at at least $\widehat{\nu^*}_x \geq \frac{2}{3}\nu^*_{(k+1)} \geq \frac{2}{3} \cdot \frac{3}{4}\widehat{\nu^*}_{(k+1)} = \frac{1}{2}\widehat{\nu^*}_{(k+1)}$. Therefore, if we store all keys $x$ with $\widehat{\nu^*}_x \geq \frac{1}{2}\widehat{\nu^*}_{(k+1)}$ we store the top-$(k+1)$ keys by $\nu^*_x$. □

A second optimization allows us to extract a larger effective sample from the sketch with $k' \geq k$ keys. This can be done when we can certify that the top-$k'$ keys by $\boldsymbol{\nu}^*$ in the transformed data are stored in the sketch $T$. Using a larger sample is helpful as it can only improve (in expectation) estimation quality (see e.g., [28, 31]). To extend this, we compute the uniform error bound $\nu^*_{(k+1)}/3$ (available because the top-$(k+1)$ keys by $\nu^*$ are stored). Let $L \leftarrow \min_{x \in T} \widehat{\nu^*}_x$. We obtain that any key $x$ in the dataset with $\nu^*_x \geq L + \nu^*_{(k+1)}/3$ must be stored in $T$. Our final sample contains these keys with the minimum $\nu^*_x$ in the set used as the threshold $\tau$.

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

# A   Properties of rHH sketches

**Sketches for $\ell_1$ heavy hitters on datasets with positive values:**   These include the deterministic counter-based Misra Gries [60, 1], Lossy Counting [56], and Space Saving [58] and the randomized Count-Min Sketch [35]. A sketch of size $O(\varepsilon^{-1})$ provides frequency estimates with absolute error at most $\varepsilon\|\boldsymbol{\nu}\|_1$. Berinde et al. [10] provide a counter-based sketch of size $O(k/\psi)$ that provides absolute error at most $\frac{\psi}{k}\|\mathsf{tail}_k(\boldsymbol{\nu})\|_1$.

**Sketches for $\ell_2$ heavy hitters on datasets with signed values:**   Pioneered by `CountSketch` [15]: A sketch of size $O(\varepsilon^{-1}\log\frac{n}{\delta})$ provides with confidence $1-\delta$ estimates with error bound $\varepsilon\|\boldsymbol{\nu}\|_2^2$ for squared frequencies. For rHH, a `CountSketch` of size $O(\frac{k}{\psi}\log\frac{n}{\delta})$ provides estimates for all squared frequencies with absolute error at most $\frac{\psi}{k}\|\mathsf{tail}_k(\boldsymbol{\nu})\|_2^2$. These bounds were further refined in [52] for $\ell_p$ rHH. The dependence on $\log n$ was replaced by $1/\psi$ in [11] for insertion only streams. Unlike the case for counter-based sketches, the estimates produced by `CountSketch` are unbiased, a helpful property for estimation tasks.

**Obtaining the rHH keys**   Keys can be arbitrary strings (search queries, URLs, terms) or integers from a domain $[n]$ (parameters in an ML model). Keys in the form of strings can be handled by hashing them to a domain $[n]$ but we discuss applications that require the sketch to return rHH keys in their string form. Counter-based sketches store explicitly $O(k/\psi)$ keys. The stored keys can be arbitrary strings. The estimates are positive for stored keys and 0 for other keys. The rHH keys are contained in the stored keys. The randomized rHH sketches (`CountSketch` and `CountMin`) are designed for keys that are integers in $[n]$. The bare sketch does not explicitly store keys. The rHH keys can be recovered by enumerating over $[n]$ and retaining the keys with largest estimates. Alternatively, when streaming (performing one sequential pass over elements) we can maintain an auxiliary structure that holds key strings with current top-$k$ estimates [15].

With general composable sketches, key strings can be handled using an auxiliary structure that increases the sketch size by a factor linear in string length. This is inefficient compared with sketches that natively store the string. Alternatively, a two-pass method can be used with the first pass computing an rHH sketch for a hashed numeric representation and a second pass is used to obtain the key strings of hashed representations with high estimates.

**Recovering (approximate) frequencies of rHH keys**  For our application, we would need to have approximate or exact frequencies of rHH keys. The estimates provided by a $(k, \psi)$ rHH sketch provide absolute error (statistical) guarantees (see Table 1). One approach is to recover exact frequencies in a second pass. We can also obtain more accurate estimates (of relative error at most $\epsilon$) by using a $(k, \epsilon\psi)$ rHH sketch.

**Testing for failure**  Recall that the dataset may not have $(k, \psi)$ rHH. We can test if one of the $k$ largest estimated frequencies to the $p$th power is below the specified error bound of $\geq \frac{\psi}{k}\|\mathsf{tail}_k(\boldsymbol{\nu})\|_p^p$. If so, we declare "failure."

# B   Overview of the proof of Theorem 3.1

For a vector $\boldsymbol{w} \in \Re^n$, permutation $\pi \in S^n$, and $p > 0$, let the random variable $\boldsymbol{w}^* \sim p\text{-ppswor}[\boldsymbol{w}] \mid \pi$ be a $p$-ppswor transform (4) of $\boldsymbol{w}$ conditioned on the event $\mathsf{order}(\boldsymbol{w}^*) = \pi$. For $q > p$ and $k > 1$, we define the following distribution:

$$F_{\boldsymbol{w},p,q,k} \mid \pi := \frac{\|\mathsf{tail}_k(\boldsymbol{w}^*)\|_q^q}{(w_{(k)}^*)^q} \ . \tag{18}$$

Note that for any $\boldsymbol{w} \in \Re^n$ and $\pi \in S^n$,

$$\Pr_{\boldsymbol{w}^*\sim p\text{-ppswor}[\boldsymbol{w}]\mid\mathsf{order}(\boldsymbol{w}^*)=\pi}\left[k\frac{|w_{(k)}^*|^q}{\|\mathsf{tail}_k(\boldsymbol{w}^*)\|_q^q} \leq \psi\right] = \Pr_{z\sim F_{\boldsymbol{w},p,q,k}\mid\pi}\left[z \leq \frac{k}{\psi}\right] \tag{19}$$

Therefore tail bounds on $F$ that hold for any $\boldsymbol{w} \in \Re^n$ and $\pi \in S^n$ can be used to establish the claim.

We now define another distribution that does not depend on $\boldsymbol{w}$ and $\pi$:

**Definition B.1.** *For $1 \leq k \leq n$ and $\rho \geq 1$ we define a distribution $R_{n,k,\rho}$ as follows.*

$$R_{k,n,\rho} := \sum_{i=k+1}^{n} \frac{\left(\sum_{j=1}^{k} Z_j\right)^\rho}{\left(\sum_{j=1}^{i} Z_j\right)^\rho} \ ,$$

*where $Z_i \sim \mathsf{Exp}[1]$ $i \in [n]$ are i.i.d.*

The proof of Theorem 3.1 will follow using the following two components:

(i) We show (Section C) that for any $\boldsymbol{w} \in \Re^n$ and permutation $\pi \in S^n$,

$$F_{\boldsymbol{w},p,q,k}|\pi \preceq R_{k,n,\rho=(q/p)} \ ,$$

where the relation $\preceq$ corresponds here to statistical domination of distributions.

(ii) We establish (Section D) tail bounds on $R_{k,n,\rho=(q/p)}$.

Because of domination, the tails bounds on $R_{k,n,\rho=(q/p)}$ provide corresponding tail bound for $F_{\boldsymbol{w},p,q,k}|\pi$ for any $\boldsymbol{w} \in \Re^n$ and $\pi \in S^n$. Together with (19), we use the tail bounds to conclude the proof of Theorem 3.1.

Moreover, the domination relation is tight in the sense that for some $\boldsymbol{w}$ and $\pi$, $F_{\boldsymbol{w},p,q,k}|\pi$ is very close to $R_{k,n,q/p}$: For distributions with $k$ keys with relative frequency $\epsilon$ and $\pi$ that has these keys in the first $k$ (as $\epsilon \to 0$), or for uniform distributions with $n \gg k$, $F_{\boldsymbol{w},p,q,k}|\pi$ (as $n$ grows).

The tail bounds (and hence the claim of Theorem 3.1) also hold without the condition on $\pi$.

**Lemma B.2.** *The tail bounds also hold for the unconditional distribution $F_{\boldsymbol{w},p,q,k}$.*

*Proof.* The distribution $F_{\boldsymbol{w},p,q,k}$ is a convex combination of distributions $F_{\boldsymbol{w},p,q,k}|\pi$. Specifically, for each permutation $\pi$ let $p_\pi$ be the probability that we obtain this permutation with successive weighted sampling with replacement. Then

$$F_{\boldsymbol{w},p,q,k} = \sum_{\pi} p_\pi F_{\boldsymbol{w},p,q,k}|\pi \ . \tag{20}$$

Since tail bounds hold for each term, they hold for the combination. $\qquad\square$

## B.1 Approximating $\psi$ by simulations

$\Psi_{k,n,\rho}(\delta)$ is the solution of the following for $\psi$:

$$\Pr_{z \sim R_{k,n,\rho}}[z \geq k/\psi] = \delta . \tag{21}$$

We can approximate $\Psi_{k,n,\rho}(\delta)$ by computing i.i.d. $z_i \sim R_{k,n,\rho}$, taking the $(1 - \delta)$ quantile $z'$ in the empirical distribution and returning $k/z'$.

From simulations we obtain that for $\delta = 0.01$ and $\rho \in \{1, 2\}$, $C = 2$ suffices for sample size $k \geq 10$, $C = 1.4$ suffices for $k \geq 100$, and $C = 1.1$ suffices for $k \geq 1000$.

## C Domination of the ratio distribution

**Lemma C.1** (Domination). *For any permutation $\pi$, $\boldsymbol{w}$, $p$, $q \geq p$, and $k \geq 1$, the distribution $F_{\boldsymbol{w},p,q,k}|_\pi$ (18) is dominated by $R_{n,k,q/p}$. That is, for all $z \geq 0$,*

$$\Pr_{z \sim F_{\boldsymbol{w},p,q,k}|\pi}\left[z \leq \frac{k}{\psi}\right] \geq \Pr_{z \sim R_{n,k,q/p}}\left[z \leq \frac{k}{\psi}\right] \tag{22}$$

*Proof.* Assume without loss of generality that $\text{order}(\boldsymbol{w}) = \pi$. Let $\boldsymbol{w}^* \sim p\text{-ppswor}[\boldsymbol{w}] \mid \text{order}(\boldsymbol{w}^*) = \pi$. Note by definition $\boldsymbol{w}^*$ is in decreasing order of magnitude. Define the random variable $\boldsymbol{y} := \boldsymbol{w}^{*p}$. $\boldsymbol{y}$ are transformed weights of a ppswor sample of $\boldsymbol{w}$ conditioned on the order $\pi$. We use use properties of the exponential distribution (see a similar use in [26]) to express the joint distribution of $\{y_i\}$. We use the following set of independent random variables:

$$X_i \sim \text{Exp}[\sum_{j=i}^{n} w_j^p] .$$

We have:

$$y_i = \frac{1}{\left(\sum_{j=1}^{i} X_i\right)^{q/p}} . \tag{23}$$

To see this, recall that $y_1$ is the (inverse) of the minimum of exponential random variables with parameters $w_1, \ldots, w_n$ and thus is (the inverse of) exponential random variable with parameter equal to their sum. Therefore, $y_1 = 1/X_1$. From memorylessness, the difference between the $(i + 1)$-st smallest inverse and the $i$-th smallest is an exponential random variable with distribution $X_i$. Therefore, the $i$-th smallest inverse has the claimed distribution (23).

We are now ready to express the random variable that is the ratio (18) in terms of the independent random variables $X_i$:

$$\frac{\sum_{j=k+1}^{n} y_j}{y_k} = \frac{\sum_{i=k+1}^{n} \frac{1}{\left(\sum_{j=1}^{i} X_i\right)^{q/p}}}{\frac{1}{\left(\sum_{j=1}^{k} X_j\right)^{q/p}}} = \sum_{i=k+1}^{n} \frac{\left(\sum_{j=1}^{k} X_j\right)^{q/p}}{\left(\sum_{j=1}^{i} X_j\right)^{q/p}} . \tag{24}$$

We rewrite this using i.i.d. random variables $Z_i \sim \text{Exp}[1]$, recalling that for any $w$, $\text{Exp}[w]$ is the same as $\text{Exp}[1]/w$. Then we have $X_i = Z_i / \sum_{j=i}^{n} w_j^p$.

We next provide a simpler distribution that dominates the distribution of the ratio. Let $W' := \sum_{j=k}^{n} w_j^p$ and consider the i.i.d. random variables $X_i' = Z_i / W'$. Note that $X_j \leq X_j'$ for $j \leq k$ and $X_j \geq X_j'$ for $j \geq k$. Thus, for $i \geq k + 1$,

$$\frac{\sum_{j=1}^{k} X_j}{\sum_{j=1}^{i} X_j} = \frac{1}{1 + \frac{\sum_{j=k+1}^{i} X_j}{\sum_{j=1}^{k} X_j}} \geq \frac{1}{1 + \frac{\sum_{j=k+1}^{i} X_j'}{\sum_{j=1}^{k} X_j'}} = \frac{\sum_{j=1}^{k} X_j'}{\sum_{j=1}^{i} X_j'} = \frac{\sum_{j=1}^{k} Z_j}{\sum_{j=1}^{i} Z_j} \tag{25}$$

This holds in particular for each term in the RHS of (24). Therefore we obtain

$$\frac{\sum_{j=k+1}^{n} y_j}{y_k} \geq \sum_{i=k+1}^{n} \frac{\left(\sum_{j=1}^{k} Z_j\right)^{q/p}}{\left(\sum_{j=1}^{i} Z_j\right)^{q/p}} .$$

$\square$

# D    Tail bounds on $R_{k,n,\rho}$

We establish the following upper tail bounds on the distribution $R_{n,k,\rho}$:

**Theorem D.1** (Concentration of $R_{n,k,\rho}$). *There is a constant $C$, such that for any $n, k, \rho$*

$$\rho = 1: \quad \Pr_{r \sim R_{n,k,\rho}}\left[r \geq Ck\ln(\tfrac{n}{k})\right] \leq 3e^{-k} \tag{26}$$

$$\rho > 1: \quad \Pr_{r \sim R_{n,k,\rho}}\left[r \geq Ck\frac{1}{\rho-1}\right] \leq 3e^{-k} \tag{27}$$

We start with a "back of the envelope" calculation to provide intuition: replace the random variables $Z_i$ in $R_{n,k,\rho}$ (see Definition B.1) by their expectation $\mathsf{E}[Z_i] = 1$ to obtain

$$S_{n,k,\rho} := \sum_{i=k+1}^{n} \frac{k^\rho}{i^\rho} \ .$$

For $\rho = 1$, $S_{n,k,\rho} \leq k(H_n - H_k) \approx k\ln(n/k)$. For $\rho > 1$ we have $S_{n,k,\rho} \approx \frac{k}{\rho-1}$. We will see that we can expect the sums not to deviate too far from this value.

The sum of $\ell$ i.i.d. $\mathsf{Exp}[1]$ random variables generates an Erlang distribution $\mathsf{Erlang}[\ell, 1]$ (rate parameter 1). The expectation is $\mathsf{E}_{r \sim \mathsf{Erlang}[\ell,1]} = \ell$ and variance is $\mathsf{Var}_{r \sim \mathsf{Erlang}[\ell,1]}[r] = \ell$. We will use the following Erlang tail bounds [48]:

**Lemma D.1.** *For $X \sim \mathsf{Erlang}[\ell, 1]$*

$$\varepsilon \geq 1: \quad \Pr[x \geq \varepsilon\ell] \leq \tfrac{1}{\varepsilon}e^{-\ell(\varepsilon-1-\ln\varepsilon)} \leq e^{1-\varepsilon}$$

$$\varepsilon \leq 1: \quad \Pr[x \leq \varepsilon\ell] \leq e^{-\ell(\varepsilon-1-\ln\varepsilon)}$$

When $\varepsilon < 0.159$ or $\varepsilon > 3.2$ we have the bound $e^{-\ell}$. For $\varepsilon > 3.2$ we also have $\tfrac{1}{\varepsilon}e^{-\ell(\varepsilon-2.2)}$

*Proof of Theorem D.1.* We bound the probability of a "bad event" which we define as the numerator being "too large" and denominators being too "small." More formally, the numerator is the sum $N = \sum_{i=1}^{k} Z_i$ and we define a bad event as $N \geq 3.2k$. Substituting $\varepsilon = 3.2$ and $\ell = k$ in the upper tail bounds from Lemma D.1, we have that the probability of this bad event is bounded by

$$\Pr_{r \sim \mathsf{Erlang}[k,1]}[r > k\varepsilon] \leq e^{-k} \ . \tag{28}$$

The denominators are prefix sums of of the sequence of random variables. We consider a partition the sequence $Z_{k+1}, \ldots, Z_n$ to consecutive stretches of size

$$\ell_h := 2^h k, (h \geq 1) \ .$$

We denote by $S_h$ the sum of stretch $h$. Note that $S_h \sim \mathsf{Erlang}[\ell_h, 1]$ are independent random variables. We define a bad event as the probability that for some $h \geq 1$, $S_h \leq 0.15\ell_h = 0.14 \, 2^h k$. From the lower tail bound of Lemma D.1, we have

$$\Pr[S_h \leq 0.15\ell_h] = \Pr_{r \sim \mathsf{Erlang}[\ell_h,1]}[r < 0.15\ell_h] \leq e^{-\ell_h} \leq e^{-2^h k} \ . \tag{29}$$

The combined probability of the union of these bad events (for the numerator and all stretches) is at most $e^{-k} + \sum_{h \geq 1} e^{-2^h k} \leq 3e^{-k}$.

We are now ready to compute probabilistic upper bound on the ratios when there is no bad event

$$
\begin{aligned}
R_{n,k,\rho} &\leq \sum_{h \geq 1} \ell_h \frac{N^\rho}{(N + \sum_{i<h} S_i)^\rho} \\
&\leq \sum_{h \geq 1} \ell_h \frac{(3.2k)^\rho}{(3.2k + 0.15\sum_{i<h}\ell_i)^\rho} \\
&= 2k\frac{(3.2k)^\rho}{(3.2k)^\rho} + \sum_{h \geq 2} 2^h k \frac{(3.2k)^\rho}{(3.2k + (2^h - 2)k)^\rho} \\
&\leq k(2 + \sum_{h=2}^{\lceil \log_2(n/k) \rceil} 2^h \left(\frac{3.2}{2^h + 1.2}\right)^\rho \leq k\left(2 + 3.2^\rho \sum_{h=2}^{\lceil \log_2(n/k) \rceil} 2^{-h(\rho-1)}\right)
\end{aligned}
$$

For $\rho = 1$ we have $O(k \log n)$. For $\rho > 1$, we have $O(k/(\rho - 1))$. $\hfill\square$

From the proof of Theorem D.1 we obtain:

**Corollary D.2.** *There is a constant $B$ such that when there are no "bad events" in the sense of the proof of Theorem D.1,*

$$\frac{\sum_{i=1}^{k} Z_i}{\sum_{i=1}^{Bk} Z_i} \leq 1/3 \ .$$

*Proof.* With no bad events, $N = \sum_{i=1}^{k} Z_i < 3.2k$ and $\sum_{i=k+1}^{k2^h} Z_i \geq 0.15k(2^h - 1)$. Solving for $0.15kB \geq 6.4k$ (for $B = 2^h - 1$ for some $h$) we obtain $B = 63$. $\hfill\square$

# E    Ratio of magnitudes of transformed weights

For $k_2 > k_1$ we consider the distribution of the ratio between the $k_2^{th}$ and $k_1^{th}$ transformed weights:

$$G_{\boldsymbol{w},p,q,k_1,k_2} \mid \pi := \left| \frac{w_{(k_2)}^{*p}}{w_{(k_1)}^{*p}} \right|$$

**Lemma E.1.** *For any $\boldsymbol{w} \in \Re^n$, $\pi \in S^n$, and $k_1 < k_2 \leq n$, the distribution $G_{\boldsymbol{w},p,q,k_1,k_2} \mid \pi$ is dominated by*

$$G'_{\rho=q/p,k_1,k_2} := \left( \frac{\sum_{i=1}^{k_1} Z_i}{\sum_{i=1}^{k_2} Z_i} \right)^{\rho} \ , \tag{30}$$

*where $Z_i \sim \mathsf{Exp}[1]$ are i.i.d.*

*Proof.* Following the notation in the proof of Lemma C.1, the distribution $G_{\boldsymbol{w},p,q,k_1,k_2}$ can be expressed as

$$\left( \frac{\sum_{i=1}^{k_1} X_i}{\sum_{i=1}^{k_2} X_i} \right)^{\rho}$$

where $X_i := \frac{Z_i}{\sum_{j=i}^{n} w_j^p}$.

For $i \in [n]$ we define $X_i' := \frac{Z_i}{\sum_{j=k_1}^{n} w_j^p}$. Now note that $X_i' \geq X_i$ for $i \leq k_1$ and $X_i' \geq X_i$ for $i \geq k_1$. Therefore,

$$\frac{\sum_{i=1}^{k_1} X_i}{\sum_{i=1}^{k_2} X_i} = \frac{1}{1 + \frac{\sum_{i=k_1+1}^{k_2} X_i}{\sum_{i=1}^{k_1} X_i}}$$

$$\leq \frac{1}{1 + \frac{\sum_{i=k_1+1}^{k_2} X_i'}{\sum_{i=1}^{k_1} X_i'}} = \frac{1}{1 + \frac{\sum_{i=k_1+1}^{k_2} Z_i}{\sum_{i=1}^{k_1} Z_i}} = \frac{\sum_{i=1}^{k_1} Z_i}{\sum_{i=1}^{k_2} Z_i} \ .$$

$\hfill\square$

# F    1-pass with total variation distance on sample $k$-tuple: upper and lower bounds

Perfect ppswor returns each subset of $k$ keys $S = \{i_1, \ldots, i_k\}$ with a certain probability:

$$p(S) = \sum_{\pi \mid \{\pi_1, \ldots, \pi_k\} = S} \prod_{j=1}^{k} \frac{w_{i_j}}{\|\boldsymbol{w}\|_1 - \sum_{h<j} w_{i_h}} \ .$$

Recall that the distribution is equivalent to successive weighted sampling without replacement. It is also equivalent to successive sampling with replacement if we "skip" repetitions until we obtain $k$ distinct keys.

With $p$-ppswor and unaggregated data, this is with respect to $\nu_x^p$. The WORp 1-pass method returns an approximate $p$-ppswor sample in terms of estimation quality and per-key inclusion probability but the TV distance on $k$-tuples can be large.

We present here another 1-pass method that returns a $k$-tuple with a polynomially small VT distance from $p$-ppswor.

---

**Algorithm 2:** 1-pass Low Variation Distance Sampling

---

**Input:** $\ell_p$ rHH method, perfect $\ell_p$-single sampler method, sample size $k$, $p$, $\delta$, $n$,

**Initialization:**

    Initialize $r = C \cdot k \log n$ independent perfect $\ell_p$-single sampling algorithms $A^1, \ldots, A^r$.

    Initialize an $\ell_p$ rHH method $R$.

**Pass 1:**

    Feed each stream update into $A^1, \ldots, A^r$ as well as into $R$.

**Produce sample:**

    $S \leftarrow \emptyset$

    For $i = 1, \ldots, r$

      Let $Out_i$ be the index returned by $A^i$

      If $Out_i \notin S$, then

        $S \leftarrow S \cup \{Out_i\}$

        For each $j > i$, feed the update $x_{Out_i} \leftarrow x_{Out_i} - R(Out_i)$ into $A^j$    `// R(Out_i) is`
`the estimate of` $x_i$ `given by` $R$

      If $|S| = k$ then exit and return $S$

    end

Output FAIL    `// Algorithms returns` $S$ `before reaching this line with high`
`probability`

---

**Theorem F.1.** *Let $p \in (0, 2]$. There is a 1-pass turnstile streaming algorithm using $k \cdot \text{poly}(\log n)$ bits of memory which, given a stream of updates to an underlying vector $x \in \{-M, -M+1, \ldots, M-1, M\}^n$, with $M = \text{poly}(n)$, outputs a set $S$ of size $k$ such that the distribution of $S$ has variation distance at most $\frac{1}{n^C}$ from the distribution of a sample without replacement of size $k$ from the distribution $\mu = (\mu_1, \ldots, \mu_n)$, where $\mu_i = \frac{|x_i|^p}{\|x\|_p^p}$, where $C > 0$ is an arbitrarily large constant.*

*Proof.* The algorithm is 1-pass and works in a turnstile stream given an $\ell_p$ rHH method and perfect $\ell_p$-single sampler methods that have this property. We use the $\ell_p$ rHH method of [52], which has this property and uses $O(k \cdot \log^2 n)$ bits of memory. We also use the perfect $\ell_p$-single sampler method of [50], which has this property and uses $\log^2 n \cdot \text{poly}(\log \log n)$ bits of memory for $0 < p < 2$ and $O(\log^3 n)$ bits of memory for $p = 2$. The perfect $\ell_p$-single sampler method of [50] can output FAIL with constant probability, but we can repeat it $C \log n$ times and output the first sample found, so that it outputs FAIL with probability at most $\frac{1}{n^C}$ for an arbitrarily large constant $C > 0$, and consequently we can assume it never outputs FAIL (by say, always outputting index 1 when FAIL occurs). This gives us the claimed $k \cdot \text{poly}(\log n)$ total bits of memory.

We next state properties of these subroutines. The $\ell_p$ rHH method we use satisfies: with probability $1 - \frac{1}{n^C}$ for an arbitrarily large constant $C > 0$, simultaneously for all $j \in [n]$, it outputs an estimate $R(j)$ for which

$$R(j) = x_i \pm \left(\frac{1}{2k}\right)^{1/p} \cdot \|\text{tail}_k(x)\|_p.$$

We assume this event occurs and add $\frac{1}{n^C}$ to our overall variation distance.

The next property concerns the perfect $\ell_p$-single samplers $A^j$ we use. Each $A^j$ returns an index $i \in \{1, 2, \ldots, n\}$ such that the distribution of $i$ has variation distance at most $\frac{1}{n^C}$ from the distribution $\mu$. Here $C > 0$ is an arbitrarily large constant of our choice.

We next analyze our procedure for producing a sample. Consider the joint distribution of $(Out_1, Out_2, \ldots, Out_{2Ck \log n})$. The algorithms $A^i$ use independent randomness. However, the

input to $A^i$ may depend on the randomness of $A^{i'}$ for $i' < i$. However, by definition, conditioned on $A^i$ not outputting $Out_{i'}$ for any $i' < i$, we have that $Out_i$ is independent of $Out_1, \ldots, Out_{i-1}$ and moreover, the distribution of $Out_i$ has variation distance $\frac{1}{n^C}$ from the distribution of a sample $s$ from $\mu$ conditioned on $s \notin \{Out_1, \ldots, Out_{i-1}\}$, for an arbitrarily large constant $C > 0$.

Let $E$ be the event that we sample $k$ distinct indices, i.e., do not output FAIL in our overall algorithm. We show below that $\Pr[E] \geq 1 - \frac{1}{n^C}$ for an arbitrarily large constant $C > 0$. Consequently, our output distribution has variation distance $1n^C$ from an idealized algorithm that samples until it has $k$ distinct values.

Consider the probability of outputting a particular ordered tuple $(i_1, \ldots, i_k)$ of $k$ distinct indices in the idealized algorithm that samples until it has $k$ distinct values. By the above, this is

$$\prod_{j=1}^{k} (1 \pm \frac{1}{n^C}) \frac{\mu_{i_j}}{1 - \sum_{j' < j} \mu_{i_{j'}}} = (1 \pm \frac{2k}{n^C}) \prod_{j=1}^{k} \frac{\mu_{i_j}}{1 - \sum_{j' < j} \mu_{i_{j'}}},$$

for an arbitrarily large constant $C > 0$. Summing up over all orderings, we obtain the probability of obtaining the sample $\{i_1, \ldots, i_k\}$ is within $(1 \pm \frac{1}{n^C})$ times its probability of being sampled from $\mu$ in a sample without replacement of size $k$, where $C > 0$ is a sufficiently large constant.

It remains to show $\Pr[E] \leq n^{-C}$ for an arbitrarily large constant $C > 0$. Here we use that for all $j \in \{1, 2, \ldots, n\}$, $R(j) = x_i \pm \left(\frac{1}{2k}\right)^{1/p} \cdot \|\text{tail}_k(x)\|_p$. Let $Y_i$ be the number of trials until (and including the time) we sample the $i$-th distinct item, given that we have just sampled $i - 1$ distinct items. The total probability mass on the items we have already sampled is at most $i \cdot \frac{1}{2k} \|\text{tail}_k(x)\|_p^p$, and thus the probability we re-sample an item already sampled is at most $\frac{1}{2}$. It follows that $\mathbf{E}[Y_i] \leq 2$. Thus, the number of trials in the algorithm is stochastically dominated by $\sum_{i=1}^{k} Z_i$, where $Z_i$ is a geometric random variable with $\mathbf{E}[Z_i] = 2$. This sum is a negative binomial random variable, and by standard tail bounds relating a sum of independent geometric random variables to binomial random variables[62] [3], is at most $Ck \log n$ with probability $1 - \frac{1}{n^C}$ for an arbitrarily large constant $C > 0$.

This completes the proof. $\qquad\square$

We now analyze the memory in Theorem F.1 more precisely. Algorithm 2 runs $r = O(k \log n)$ independent perfect $\ell_p$-sampling algorithms of [50]. The choice of $r = O(k \log n)$ is to ensure that the variation distance is at most $\frac{1}{\text{poly}(n)}$; however, with only $r = O(k)$ such samplers, the same argument as in the proof of Theorem F.1 gives variation distance at most $2^{-\Theta(k)}$. Now, each $\ell_p$-sampler of [50] uses $O(\log^2 n (\log \log n)^2)$ bits of memory for $0 < p < 2$, and uses $O(\log^3 n)$ bits of memory for $p = 2$. We also do not need to repeat the algorithm $O(\log n)$ times to create a high probability of not outputting FAIL; indeed, already if with only constant probability the algorithm does not output FAIL, we will still obtain $k$ distinct samples with $2^{-\Theta(k)}$ failure probability provided we have a large enough $r = O(k)$ number of such samplers.

Algorithm 2 also runs an $\ell_p$ rHH method, and this uses $O(k \log^2 n)$ bits of memory [52]. Consequently, to acheive variation distance at most $2^{-\Theta(k)}$, Algorithm 2 uses $O(k \log^2 n (\log \log n)^2)$ bits of memory for $0 < p < 2$, and $O(k \log^3 n)$ bits of memory for $p = 2$.

We now show that for $0 < p < 2$, the memory used of Algorithm 2 is best possible for any algorithm, up to a multiplicative $O((\log \log n)^2)$ factor. For $p = 2$, we show our algorithm's memory is optimal up to a multiplicative $O(\log n)$ factor. Further, our lower bound holds even for any algorithm with the much weaker requirement of achieving variation distance at most $\frac{1}{3}$, as opposed to the variation distance at most $2^{-\Theta(k)}$ that we achieve.

**Theorem F.2.** *Any* 1-*pass turnstile streaming algorithm which outputs a set $S$ of size $k$ such that the distribution of $S$ has variation distance at most $\frac{1}{3}$ from the distribution of a sample without replacement of size $k$ from the distribution $\mu = (\mu_1, \ldots, \mu_n)$, where $\mu_i = \frac{|x_i|^p}{\|x\|_p^p}$, requires $\Omega(k \log^2 n)$ bits of memory, provided $k < n^{C_0}$ for a certain absolute constant $C_0 > 0$.

*Proof.* We use the fact that such a sample $S$ can be used to find a constant fraction of the $\ell_q(k,1)$ residual heavy hitters in a data stream. Here we do not need to achieve residual error for our lower bound, and can instead define such indices $i$ to be those that satisfy $|x_i|^p \geq \frac{1}{k}\|x\|_p^p$. Notice that there are at most $k$ such indices $i$, and any sample $S$ (with or without replacement) with variation distance at most $1/3$ from a true sample has probability at least $1 - (1-1/k)^k - 1/3 \geq 1 - 1/e - 1/3 \geq .29$ of containing the index $i$. By repeating our algorithm $O(1)$ times, we obtain a superset of size $O(k)$ which contains any particular such index $i$ with arbitrarily large constant probability, and these $O(1)$ repetitions only increase our memory by a constant factor.

It is also well-known that there exists a point-query data structure, in particular the `CountSketch` data structure [15, 67], which only needs $O(\log|S|) = O(\log k)$ rows and thus $O((k\log k)\log n)$ bits of memory, such that given all the indices $j$ in a set $S$, one can return all items $j \in S$ for which $|x_j|^p \geq \frac{1}{k}\|x\|_p^p$ and no items $j \in S$ for which $|x_j|^p < \frac{1}{2k}\|x\|_p^p$. Here we avoid the need for $O(\log n)$ rows since we only need to union bound over correct estimates in the set $S$.

In short, the above procedure allows us to, with arbitrarily large constant probability, return a set $S$ containing a random .99 fraction of the indices $j$ for which $|x_j|^p \geq \frac{1}{k}\|x\|_p^p$, and containing no index $j$ for which $|x_j|^p < \frac{1}{2k}\|x\|_p^p$.

We now use an existing $\Omega(k\log^2 n)$ bit lower bound, which is stated for finding all the heavy hitters [52], to show an $\Omega(k\log^2 n)$ bit lower bound for the above task. This is in fact immediate from the proof of Theorem 9 of [52], which is a reduction from the one-way communication complexity of the Augmented Indexing Problem and just requires any particular heavy hitter index to be output with constant probability. In particular, our algorithm, combined with the $O((k\log k)\log n)$ bits of memory side data structure of [15] described above, achieves this.

Consequently, the memory required of any 1-pass streaming algorithm for the sampling problem is at least $\Omega(k\log^2 n) - O((k\log k)\log n)$ bits, which gives us an $\Omega(k\log^2 n)$ lower bound provided $k < n^{C_0}$ for an absolute constant $C_0 > 0$, completing the proof. □

## G  Estimates of one-pass WORp

We first review the setup. Our one-pass WORp method returns the top $k$ keys by $\widehat{\nu_x^*}$ as our sample $S$ and returns $\widehat{\nu^*}_{(k+1)}$ as the threshold. The estimate of $f(\nu_x)$ is 0 for $x \notin S$ and for $x \in S$ is

$$\widehat{f(\nu_x)} := \frac{f(\widehat{\nu_x^*}r_x^{1/p})}{1 - \exp\left(-r_x\left(\frac{\widehat{\nu_x^*}}{\widehat{\nu^*}_{(k'+1)}}\right)^p\right)} \ . \tag{31}$$

We assume that $f(w)$ is such that for some constant $c$,

$$\forall \varepsilon < 1/2, |f((1+\varepsilon)w) - f(w)| \leq c\varepsilon f(w) \ . \tag{32}$$

We need to establish that

$$\mathsf{Bias}[\widehat{f(\nu_x)}] \leq O(\varepsilon)f(\nu_x)$$
$$\mathsf{MSE}[\widehat{f(\nu_x)}] \leq (1+O(\varepsilon))\mathsf{Var}[\widehat{f(\nu_x)}'] + O(\varepsilon)f(\nu_x)^2 \ ,$$

where $\widehat{f(\nu_x)}'$ are estimates obtained with a (perfect) $p$-ppswor sample.

*Proof of Theorem 5.1.* From (10), the rHH sketch has the property that for all keys in the dataset,

$$\|\widehat{\boldsymbol{\nu}^*} - \boldsymbol{\nu}^*\|_\infty \leq \varepsilon\nu^*_{(k+1)} \ . \tag{33}$$

For sampled keys, $|\nu_x^*| \geq |\nu^*_{(k+1)}|$ and hence $|\widehat{\nu_x^*} - \nu_x^*| \leq \varepsilon|\nu^*_{(k+1)}| \leq \varepsilon|\nu_x^*|$. Using (6) we obtain that $\|\nu_x' - \nu_x\| \leq \varepsilon|\nu_x|$.

From our assumption (32) , we have $|f(\nu_x') - f(\nu_x)| \leq c\varepsilon f(\nu_x)$.

We consider the inclusion probability and frequency estimate of a particular key $x$, conditioned on fixed randomization $r_z$ of all other keys $z \neq x$. The key $x$ is included in the sample if $\widehat{\nu_x^*} \geq \widehat{\boldsymbol{\nu}^*}_{(k+1)}$. We consider the distribution of $\widehat{\nu_x^*}$ as a function of $r_x \sim \mathsf{Exp}[1]$. The value has a form of $E + \nu_x / r_x^{1/p}$, where the erro $E$ satisfies $|E| \leq \varepsilon |\nu_{(k+1)}^*|$. The conditioned inclusion probability thus falls in the range

$$p_x' = \Pr[\nu_x / r_x^{1/p} \pm \varepsilon |\nu_{(k)}^*| \geq \widehat{\boldsymbol{\nu}^*}_{(k)}] = \Pr\left[r_x \leq \left(\frac{\nu_x}{\widehat{\boldsymbol{\nu}^*}_{(k+1)} \pm \varepsilon |\nu_{(k)}^*|}\right)^p\right]$$

$$= 1 - \exp(-\left(\frac{\nu_x}{\widehat{\boldsymbol{\nu}^*}_{(k+1)} \pm \varepsilon |\nu_{(k)}^*|}\right)^p) \ .$$

We estimate $p_x'$ by

$$p_x'' = 1 - \exp\left(-r_x(\frac{\widehat{\nu_x^*}}{\widehat{\boldsymbol{\nu}^*}_{(k+1)}})^p\right) \ . \tag{34}$$

This estimate has a small relative error. This due to the relative error in $\widehat{\nu_x^*}$ and because $|(1 - \exp(-(1 \pm \epsilon)b))) - (1 - \exp(-b))| = O(\epsilon)(1 - \exp(-b))$ and $(\frac{\nu_x'}{\widehat{\boldsymbol{\nu}^*}_{(k)}})^p)$ is an $O(\epsilon)$ relative error approximation of $(\frac{\nu_x}{\widehat{\boldsymbol{\nu}^*}_{(k)} - E})^p$.

We first consider the bias. Instead of using the unbiased inverse probability estimate $f(\nu_x)/p_x'$ when $x$ is sampled (with probability $p_x'$) our estimator (17) ($f(\nu_x')/p_x''$ approximates both the numerator and the denominator.

In the numerator of the estimator, we replace $f(\nu_x)$ by the relative error approximation $f(\nu_x')$. Therefore overall, we use a small relative error estimate of the actual inverse probability estimate when it is non zero, which translates to a bias that is $O(\epsilon)f(\nu_x)$.

We next bound the Mean Squared Error (MSE) of the estimator (17). We express the variance contribution of exact $p$-ppswor conditioned on the same randomization $r_z$ of all keys $z \neq x$. This is $\mathsf{Var}[\widehat{f(\nu_x)}]' = (1/p_x - 1)f(\nu_x)^2$, where $p_x = \Pr[\nu_x/r_x^{1/p} \geq \boldsymbol{\nu}_{(k)}^*] = 1 - \exp(-\left(\frac{\nu_x}{\nu_{(k)}^*}\right)^p)$. The MSE contribution is

$$p_x'(f(\nu_x')/p_x'' - f(\nu_x))^2 + (1 - p_x')f(\nu_x)^2 \ . \tag{35}$$

We observe that the approximate threshold (that determines $p_x'$) approximates the perfect $p$-ppswor threshold: $|\widehat{\boldsymbol{\nu}^*}_{(k)} - \boldsymbol{\nu}_{(k)}^*| \leq \varepsilon |\nu_{(k)}^*|$. When $p_x < 1/2$, (35) approximates $\mathsf{Var}[\widehat{f(\nu_x)}]'$ with relative error $O(\varepsilon)$.

When $p_x$ is close to 1 this is dominated by $O(\varepsilon)f(\nu_x)^2$.

$\square$

We remark that our analysis of the error only assumed the rHH error bound (33) which holds for all sketch types including `Counters`. The bias analysis can be tightened for `CountSketch` that returns unbiased estimates of the frequency.

# H  Pseudocode

**Algorithm 3:** 2-pass WORp

**Input:** $\ell_q$ rHH method, sample size $k, p, \delta, n$,

**Initialization:**

Draw a random hash $r_x \sim \mathcal{D}$        `// Random map of keys `$x$` to `$r_x$

$\psi \leftarrow \frac{1}{3}\Psi_{n,k,q/p}(\delta)$

Initialize `KeyHash`        `// random hash function from strings to `$[n]$

$R.\texttt{Initialize}(k, \psi)$        `// Initialize rHH structure randomization`

**Pass I:**        `// Use composable aggregation (process input keys into rHH structures and merge rHH structures)`

**begin**

     **Process data element** $e = (e.\mathsf{key}, e.\mathsf{val})$ into rHH sketch $R$

     $R.\texttt{Process}(\texttt{KeyHash}(e.\mathsf{key}), e.\mathsf{val}/r_{e.\mathsf{key}}^{1/p})$      `// Generate and process output element`

**Pass II:**    `// For keys with top `$2k$` estimates `$\widehat{\nu_x^*}$`, collect exact frequencies `$\nu_x$`.`

Initialize a composable top-$2k$ structure $T$. The structure stores for each key its priority and frequency. The structure collects exact frequencies for the keys with top $2k$ priorities.

$\texttt{Merge}(T_1, T_2)$: Add up values and retain $3k$ top priority keys.

**Process data element** $e = (e.\mathsf{key}, e.\mathsf{val})$ into $T$

**begin**

     **if** $e.key \in T$ **then**

         $T[e.\mathsf{key}].val+ = e.\mathsf{val}$

     **else**

         $est \leftarrow R.\texttt{Est}[\texttt{KeyHash}(e.\mathsf{key})]$      `// `$\widehat{\nu_{\mathsf{key}}^*}$

         **if** $est >$ *lowest priority in* $T$ **then**

             Insert $e.\mathsf{key}$ to $T$

             $T[e.\mathsf{key}].val \leftarrow e.\mathsf{val}$

             $T[e.\mathsf{key}].priority \leftarrow est$

             **if** $|T|$>$2k$ **then**

                 Eject lowest priority key from $T$

**Produce sample:** Sort $T$ by $T[x].val * r_x^{1/p}$        `// actual `$\nu_x^*$` for keys in `$T$

Return $(x, T[x].val)$ for top-$k$ keys and $(k+1)$th $T[x].val * r_x^{1/p}$