[Reviews · NeurIPS 2020]

Review 1

Summary and Contributions: The paper presents an approach for weighted sampling of keys by powers of their frequency, crucially without replacement, in the context of a sequence of data in which keys may appear multiple times. The technique works by transforming the inputs, reweighting by some random noise; then sampling consists of finding the top-k elements. This can be done using off-the-shelf sketch datastructures for finding heavy hitters. The authors present multiple variations on this approach and some experiments.

Strengths: The authors make the point that their method took some cleverness to come up with and analyze, yet its actual implementation is very easy and uses only simple transformations of the input data combined with simple off-the-shelf tools (heavy hitters sketches are widely available). This seems to be the major strength of the work.

Weaknesses: Although I don't have much background in the field, it seems like the scope of the work may be sort of narrow -- weighted sampling for data that cannot fit entirely in memory, where weights are scaled by a power, without replacement, in the case where keys may have negative updates to their weights, seems like an oddly specific problem setting. The authors do make the case that all these things may have practical relevance and previous work had not addressed all of them at once.

Correctness: I was not able to evaluate the correctness of the proofs. The empirical methodology seems sort of sparse (one toy distribution), but that's probably not the point of a sketching algorithms paper.

Clarity: It is not especially clear -- it is certainly very dense with details and background facts about the algorithms used. Note I have little background in the field, so while it was hard for me to read, maybe this is to be expected.

Relation to Prior Work: Yes, there is a section on previous contributions that details various approaches to related problems. It seems like there are previous approaches for lp weighted WOR sampling that do not support negative updates for keys, and approaches for lp weighted sampling that do support negative updates but cannot do WOR. It might help to explicitly discuss which of these approaches are most similar in their implementations.

Reproducibility: Yes

Additional Feedback: A clearer outline up front about the progression from bottom k sampling to sampling by transformed frequency to picking the k keys using rHH would have been helpful, I think probably even for a better-informed audience than me. I checked "yes" on the "broader impact" question even though that section is not included, because it seems clear this is a pure theory paper where the discussion is not necessary. I didn't notice the pun in the title at first, but when I did notice it I chuckled slightly. After author response: Thanks to the authors for correcting my misunderstanding about the scope of the algorithms and relationship to previous work. After reading the reviews and author response, I find that the presentation of the paper is still very difficult to follow. These were the two main weaknesses, and my score remains the same (and is still low confidence).


Review 2

Summary and Contributions: This paper designs a new sketch for WithOut Replacement (WOR) of frequencies of keys from a multi-set stream, but proportional to the frequencies to the power p for p in [0,2]. In particular the algorithm should sample at most one item for each label in the multi-set (without replacement), since the goal is to understand the distribution of the frequencies. It can weight them to represent the frequencies. The most interesting cases are p=0 (is it there or not), p=1 (total count), and p=2 (count squared -- related to l_2 norm errors). These are all covered in the meta-algorithms presented. The algorithms are essentially: - compute a rHH sketch of data elements, with weights transformed by ^p (a rHH sketch is a heavy-hitters sketch with residual error -- all standard ones, CountMin, CountSketch, Misra-Gries) satisfy this. - produce a sample from this sketch - reweight the sampled items to account for bias/variance as possible. The details of the weight transforms and the reweighting are a bit technical, and the rational for that is derived in the Supplement. But the algorithms are quite simple.

Strengths: I like the results, I find them interesting and non-trivial.

Weaknesses: But I have some reservation based on its tenuous connection to other core topics in NeurIPS, and the following concerns. While this work has simple algorithms, the writing is a bit hard to follow since it is very notation heavy. There is a lot of background on previous works, and not all of them seem to be used, but I think contributes to the complex notation. I like that it is able to be described concretely and precisely in 8 pages, but I feel it would be more useful to introduce the simple algorithms first, and then elaborate on the analysis. Perhaps even a small algorithm block would have helped. Also, it is not clear why *samples* are needed, instead of just using the original rHH sketches? If I am understanding the error bounds, they seem weaker than the existing rHH sketches -- please clarify what is added by using samples. All experiments are for p=1 or p=2 where there are well-known, tried-and-true rHH sketches. All or almost all error from the new sketches appear to be over-estimates (Although quite small). It would be useful to see if using the MG sketch (which only under estimates) these may even balance better in case the sampling also contribute to over-estimates. Also it would be useful to also put the rHH sketches (without the added sampling sketch) as comparison in the experiments -- or to demonstrate or explain why they are not comparable.

Correctness: I did not attempt to verify the proofs, but I have no reason to doubt the correctness.

Clarity: Not really. See above.

Relation to Prior Work: The concept referred to as "composable sketches" appears to be the same as the "mergeable summaries" in this paper: https://www.researchgate.net/publication/254006519_Mergeable_Summaries which in fact demonstrated it held for the various rHH sketches discussed. Also, why not just use standard rHH sketches for this problem, what is the advantage for the most important cases of p=1 and p=2?

Reproducibility: Yes

Additional Feedback: --- after rebuttal --- My concerns about the writing persist. But the authors answered clearly my concerns about (1) fit, and (2) why sampling is better than just heavy hitters. After the explanation, I like the title now :)


Review 3

Summary and Contributions: The paper proposes a method for sampling without replacement proportionally to a transformation x -> x^p on the frequency, that reduces the computation of a quantity to a few, important elements.

Strengths: The work proposes a computationally efficient approach to efficient SWOR sampling. The analysis seems thorough, though as stated in my confidence evaluation I did not understand it in depth. Though I am familiar with sampling theory, I am not familiar with the notions introduced in this work, and thus can hardly assess its significance and novelty.

Weaknesses: I feel that the paper is not accessible enough for the NeurIPS community. Many notions are properly introduced in the middle of the paper, which makes the paper rather obscure even to audience familiar with sampling. Specifically, I refer to: - what is a negative update ? - heavy hitters are only defined in 2.3. It seems that for a specific epsilon, any element can be a heavy hitter, so referring to a l_2 heavy hitter refers to any element, - what is p ? Without context (e.g. in title, in "the case for p's) p refers to a probability but not here, \ell_p is already clearer. - in the definition of composable sketches, the expression "data structures that are such that the desired output can be produced from the sketch" is not a clear definition. --- After rebuttal --- After reading the rebuttal and the other reviews, I still believe that: while the paper might be well-considered in a data mining conference, it is not accessible enough to the general ML community. For this reason, I still believe it should be rejected. I don't believe the claim that distributed sketching appears significantly at all generalist ML conference, but in doubt I'll raise by score to weak reject.

Correctness: I think the claims and method can be trusted.

Clarity: Specific points: - The legends of both Figure 1 and 2 are unreadable. - Avoid inline fractions. - The first equality below eq 2 is useless.

Relation to Prior Work: The paper explains the relation to prior work, which seems to make its contribution clear for people that are very familiar to related work.

Reproducibility: Yes

Additional Feedback:

[Author Response · NeurIPS 2020]

**NeurIPS5595 author response: WOR and $p$'s: Sketches for $\ell_p$-Sampling Without Replacement**

We appreciate your careful feedback on our submission. We will use your concerns and suggestions to improve the
presentation to make it more accessible. Below we address the specific points raised by the reviewers.

**R1:** Though the problem setting may seem specific at first glance, we believe that the contribution is of broad interest:

• Weighted random sampling has been studied intensively for decades and applied across many disciplines.
Without-replacement sampling has also been extensively studied and used across disciplines to improve
performance on skewed data sets, so is not "fringe."

• Sampling is important in machine learning applications and optimization, for example, in learning algorithms
that use importance sampling to select training examples or features. Performing aggregations efficiently on
data that does not fit in memory (streaming, distributed, federated) is important. Data commonly lives on
multiple devices, servers, cloud components, etc.

• The powers in the range $[0, 2]$ are important, well-studied, and applied to frequencies (say of training data) in
practice. They are related to $p$-norms. When $p < 1$, high frequencies are mitigated (which is commonly done
in weighting training examples by frequency, for example, in word models). The choice $p = 1$ corresponds to
sampling by frequency. The choice $p = 2$ emphasizes larger entries. For example, we are likely to sample
the "$\sqrt{n}$" in the vector $(1, 1, 1, \ldots, 1, \sqrt{n}, 1, 1, \ldots, 1)$, whereas for $p = 1$ we are not. This can lead to lower
variance estimators for functions that are sensitive to large weights.

**R1:** Our work is novel in broader regimes than stated in the review. Support for negative updates is important, but our
work is novel also when we only have positive updates. That is, we provide the first known WOR method for $p > 1$.
Prior methods for related problems use different techniques, but we acknowledge that the paper should discuss this in
more detail. In particular, prior WOR approaches for $p \leq 1$ are efficient to implement but do not allow samples to be
coordinated because the randomization is not reproducible, and they do not support negative updates. Methods based
on random projections are asymptotically efficient and support negative updates but have large hidden constants, do not
support coordination of samples, and do not support WOR. Our more general problem required a different technique.

**R2:** The question of scope for NeurIPS/ICML is a fluid one as the conference greatly expanded in recent years.
ICML/NeurIPS programs routinely include multiple papers that present or improve fundamental tools that are building
blocks in ML implementations: streaming/distributed aggregation, sketching and sampling, matrix computations, and
generally optimizing the computation/data transfer of the training process. The program also includes methods that build
on these tools. For example, from NeurIPS 2019: (i) Communication-efficient Distributed SGD with Sketching, (ii)
Extreme Classification in Log Memory using Count-Min Sketch: A Case Study of Amazon Search with 50M Products,
and (iii) Sampling sketches for concave sublinear functions of frequencies. As for our specific contribution, weighted
sampling is a fundamental tool that applies to many aspects of ML (this is discussed briefly in the introduction).

**R2:** Concerning using other heavy hitters methods: we did a proof of concept implementation and so far only used
count-sketch, but our use of the HH sketch is indeed a black box. We agree with the reviewer that it would be nice to
implement our method with the MG sketch. This will be advantageous in the regime $p \leq 1$ and for positive updates
(this is what the MG sketch supports). In this regime it will be much more efficient than count-sketch (which we used in
our experiments). Moreover, the error guarantee is for worst-case streams, but MG should be better in practice.

**R2:** About sampling versus finding heavy hitters: an rHH sketch alone only gives us heavy hitter keys but no information
on the tail. Some datasets may not have heavy hitter keys. A weighted sample exposes the respective HH (if there are
any) but is much more powerful. In particular, we can obtain unbiased estimates of sum queries and domain queries. For
example, to estimate the total "weight" (sum of powers of frequencies) of keys that are not heavy, obtain an unbiased
gradient update estimate, or estimate the value of a loss function (or other function of the form of a sum over keys).
An HH sketch cannot provide that in general. Our careful scaling approach obtains a weighted random sample by
(essentially) turning "random" keys into HH with probability proportional to a power of their frequency. A sample can
also be more interpretable than a sketch, providing us representative examples of a dataset.

**R3:** About the exposition: we will work to make the introduction more accessible. The general area of the paper is
streaming/distributed sketching/sampling. We know this community uses a somewhat different language than, say, a
probability theorist. However, this is a broad area that is fundamental to efficient computation and with ML applications.
In each ML conference there are several papers that use or develop sketching techniques. Our use of $p$ in the context of
$\ell_p$ norms/sampling/HH is standard in that community. The title is meant to allude to the book "War and Peace". Our
apologies for the confusion, this can be addressed.

[Meta-Review · NeurIPS 2020]

While the results are promising, the writing can be clarified and improved for the final version.